# Beta-Thalassemia Minor and SARS-CoV-2: Physiopathology, Prevalence, Severity, Morbidity, and Mortality

**Edouard Lansiaux** [1,*] , **Emmanuel Drouin** [2,*] and **Carsten Bolm** [3,*]

1 Henry Warembourg School of Medicine, Lille University, 2 Avenue Eugène Avinée, 59120 Loos, France
2 Neurology Service, Lille Catholic Institute Hospital Group, GHICL (Groupe Hospitalier de l'Institut Catholique de Lille), 115 Rue du Grand But, 59462 Lomme, France
3 Institute of Organic Chemistry, RWTH Aachen University, Landoltweg 1, D-52074 Aachen, Germany
* Correspondence: edouard.lansiaux.etu@univ-lille.fr (E.L.); emmanuel.drouin@univ-catholille.fr (E.D.); carsten.bolm@oc.rwth-aachen.de (C.B.)

**Abstract:** Background: Since the first year of the COVID-19 global pandemic, a hypothesis concerning the possible protection/immunity of beta-thalassemia carriers has remained in abeyance. Methods: Three databases (Pubmed Central, Scopus, and Google Scholar) were screened and checked in order to extract all studies about the incidence of confirmed COVID-19 cases, mortality rate, severity assessment, or ICU admission among patients with beta-thalassemia minor, were included in this analysis. The language was limited to English. Studies such as case reports, review studies, and studies that did not have complete data for calculating incidences were excluded. Results and discussion: a total of 3 studies out of 2265 were selected. According to our systematic-review meta-analysis, beta-thalassemia carriers could be less affected by COVID-19 than the general population [IRR = 0.9250 (0.5752; 1.4877)], affected by COVID-19 with a worst severity [OR = 1.5933 (0.4884; 5.1981)], less admissible into the ICU [IRR = 0.3620 (0.0025; 51.6821)], and more susceptible to die from COVID-19 or one of its consequences [IRR = 1.8542 (0.7819; 4.3970)]. However, all of those results remain insignificant with a bad *p*-value (respectively 0.7479, 0.4400, 0.6881, and 0.1610). Other large case-control or registry studies are needed to confirm these trends.

**Keywords:** COVID-19; SARS-CoV-2; respiratory distress; beta thalassemia; minor; immunization; heme

## 1. Introduction

The world has had to face a COVID-19 pandemic for two years, followed by successive waves of SARS-CoV-2 variants. This infection was first discovered in December 2019 in the South China Seafood Market, Hubei Province, China [1]. Since the pandemic's first months, an epidemiological particularity has been observed concerning minor beta-thalassemia patients: they seem to be protected against severe COVID-19 forms, and in this way, the beta-thalassemic trait would confer a certain immunization against COVID-19 [2,3]. A few times after, a scientific niche debate (ed. between experts) was opened on the question: some epidemiological datasets supported this hypothesis [4–7], despite the fact that in vitro experiments tend to demonstrate a lack of a mechanistic link between Hb variants and disease effects [8]. This hypothesis is currently being developed on the physiopathological side and is based upon three main ways: iron, heme, and redox metabolism; erythropoietin and the ACE cascade; and the BCL11A gene. In the first part of our letter, we are going to develop them; in the second, a meta-analysis concerning epidemiological trackers was carried out in order to study this hypothesis.

## 2. Physiopathology

### 2.1. Beta-Thalassemia Minor

Beta-thalassemia is a heterozygous disease caused by a genetic point mutation on a single allele of the β globin heme on chromosome 11. Therefore, contrary to the normal adult hemoglobin (96% HbA with $\alpha\alpha\beta\beta$ composition, 2% $HbA_2$ with $\alpha\alpha\Delta\Delta$ composition, and 2% HbF with $\alpha\alpha\gamma\gamma$ composition), patients with beta-thalassemia minor present a special HbA composition with $\alpha\alpha\beta\beta^e$ or $\alpha\alpha\beta\beta^+$ model. In this way, they suffer from a mild decrease in β-chain production relative to α-chain and, thus a decrease in β-chain available for HbA synthesis. In order to compensate for this default, the human system increases the production of non-affected globins such as HbA2 and HbF [4]. Due to that, a lot of biological cascades are put into play: iron metabolism (due to the iron excess caused by abnormally shaped erythrocytes destruction), ineffective erythropoiesis, and free α-chains precipitations . . .

### 2.2. Iron, Heme, and Redox

The transcribed, non-structural SARS-CoV-2 proteins ORF 8, 3a, and 10 play critical roles during SARS-CoV-2 replication and COVID-19 pathogenesis (by activating NF-kB-mediated inflammation and immune responses) [9]. The initial hypothesis (i.e., beta-thalassemia minor protection against COVID-19) was based upon a preprint that showed that SARS-CoV-2 ORF8 and surface glycoproteins could gather with the porphyrin in order to form a complex. Meanwhile, ORF1ab, ORF10, and ORF3a proteins could dissociate the iron ions from the heme (viral proteins may interact with α-chains via the Isd domains and then break β-chains using the IgA1 protease structure) by attacking the heme (porphyrin, which is initially formed in the mitochondria) on the hemoglobin 1-beta chain [9].

Beta-thalassemia patients involve three cell protection systems during erythropoiesis.

In β-thalassemia, iron control (and therefore the control of heme) is a priority for cell survival in order to avoid the overload or the proteotoxicity caused by iron deficiency. In this last scenario (lack of iron), the heme-regulated inhibitor of protein translation (HRI) allows to inhibit globin translation into heme-deficient erythroid precursors [10]; that is thanks to the heme-regulated eukaryotic initiation factor-2α (eIF2α) kinase, which phosphorylates eIF2 subunit. According to in vitro studies, the activation of HRI (with ROS implication) needs the presence of heat shock proteins 70 and 90 [11]. An in vivo experiment (beta-thalassemic animal models with a genetic deficiency of HRI have shown a more severe hematological phenotype than normal β-thalassemia ones) supports the main role of eIF2α in stress erythropoiesis [12]. The activating transcription factor 4 (ATF4) mRNA translational up-regulation by the HRI-eIF2αP signaling pathway was required to soften proteotoxicity and support the differentiation of erythroids [13] (Figure 1A). Therefore, the HRI limited proteotoxicity and permitted ATF4 protein expression in order to keep the main mitochondrial functions and oxidative homeostasis [14] (Figure 1A).

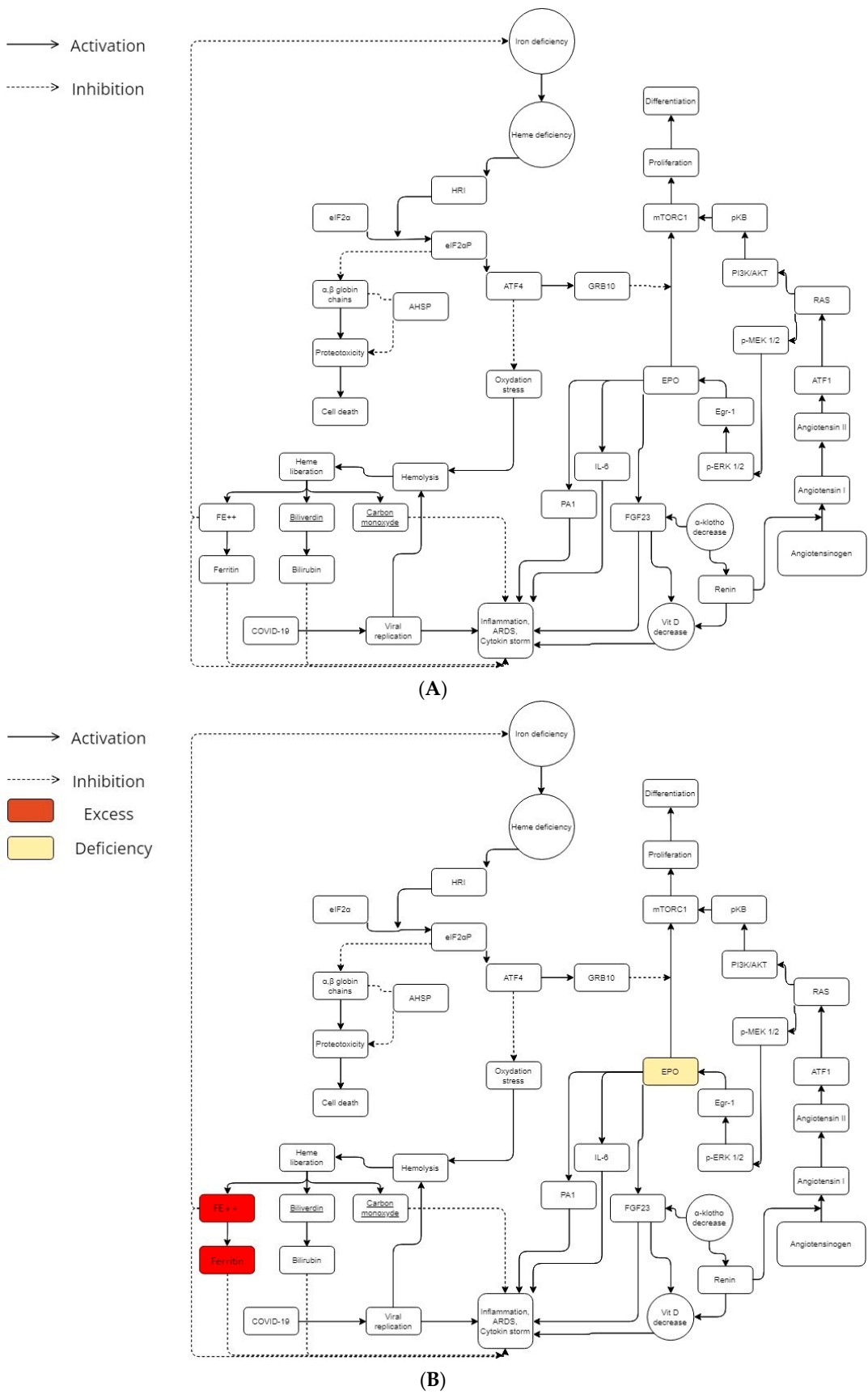

**Figure 1.** (**A**) Iron, angiotensin, and erythropoietin cascades during COVID-19 infection in a normal

metabolism. In the general population, metabolic balance leads to a more or less severe possible COVID-19 infection. (**B**) Iron, angiotensin, and erythropoietin cascades during COVID-19 infection in a beta-thalassemic carrier metabolism. Beta-thalassemia minor patients have low EPO levels. Furthermore, they have an excess of iron and ferritin. In his way, beta-thalassemia carriers have a different metabolic balance, which could explain the mild or less severe COVID-19 infection in this population.

Heme consumption has a certain importance in erythropoiesis (and especially when you have an overload of iron due to beta-thalassemia). HO-1 (heme oxygenase-1) is a protein that catalyzes heme catabolism [15]. It is generally considered a protective enzyme due to its pro-oxidant "free" heme breakdown ability and release of antioxidants such as biliverdin and bilirubin (Figure 1A,B). Several studies demonstrated that some HO-1 gene polymorphisms, especially the promoter region GT dinucleotide repeat mutation, controls the inducibility of HO-1 to ROS [16–21]. According to them, GT sequences with short alleles are associated with HO-1 inducibility, resulting in amplified cytoprotection [21]. COVID-19 complications may cause patients to have longer GT sequences and decreased vessel hemostasis.

Among atypical chaperone systems, the α hemoglobin stabilizing protein (AHSP) plays an important role in beta-thalassemic erythropoiesis. AHSP coheres to α-Hb, averts its precipitation (which induces splenomegaly into the beta-thalassemia minor [9]), and avoids free α-Hb toxicities. The α-Hb attached to AHSP is stronger to the precipitation induced by oxydation [22]. In addition to that, AHSP knock-out animal models displayed semiologic features and an ineffective degree of erythropoiesis related to β-thalassemia [23].

In view of the facts that beta-thalassemia subjects (and more particularly beta-thalassemia minor) do not seem to have a porphyrin deficiency [24,25], and ferritin and iron excess appear to be a risk factor for COVID-19 exacerbation [26,27], the possible protection/immunization of beta-thalassemic minor patients could be attributed to another physiopathological mechanism than the heme one: the absence of the ORFs interaction with beta-chain hemoglobin (due to its absence in beta-thalassemia). Those beta-thalassemia protective mechanisms previously described might allow to keep a normal rate of porphyrins, iron, and ferritin during COVID-19 infection (Figure 1B). It may even be that the latter contribute to reducing the cytokine storm by maintaining redox homeostasis.

### 2.3. Erythropoietin and ACE Way

The symptoms of altitude sickness at high altitude are similar to those of COVID-19 [28,29]. Erythropoietin (EPO) seems to be the common denominator between COVID-19, beta-thalassemia, and altitude sickness [30]. It has been hypothesized that modulations of the host immune system induced by malarial selection pressure thanks to thalassemia/HbE mutations might give this protection as an antimalarial effect [31]. The I/D polymorphism of the D allele of angiotensin converting enzyme (ACE) is significantly linked to mild malaria vs. severe malaria, which increases ACE levels and subsequently increases angiotensin II (Ang II) production compared to the I allele [32,33]. Therefore, a common denominator needs to be traced in order to demonstrate the two genetic determinants emergence forced by malarial evolutionary pressure. Both genetic determinants (evolutionary thalassemias selection and the ACE D allele) appear to elicit and sustain a potential phylogenetically preserved ancestral protective innate immune response mechanism against pathogen invasion mediated, either due to systemic or/and local increases in erythropoietin (EPO) production [34]. In this way, beta-thalassemic patients have low EPO levels according to the degree of anemia [35,36] (Figure 1B). Furthermore, concerning the ACE, the increase of activating transcription factor 1 (ATF1) may be an important key to the severe COVID-19 explanation [37].

### 2.4. BCL11A Gene Possible Imputation

The broad variation in COVID-19 infection rates and its severity could potentially be explained by genetic variability between human and animal hosts. In this way, the

observed protection/immunization of beta-thalassemia minor against COVID-19 could have a genetic origin: B-cell lymphoma 11 A (BCL11A) gene which was identified thanks to genome-wide association studies (GWAS) as an associated gene with fetal hemoglobin (HbF) production [38] and therefore as a modulator of beta-thalassemia severity [39–41].

In this way, we have described the three main mechanisms that support our initial hypothesis; unfortunately, there are no real in silico/in vivo/in vitro studies that demonstrate that. However, there are also epidemiological studies that support this hypothesis [42,43] and there is no evidence of a significant relationship between COVID-19 infection and the oxyhemoglobin dissociation curve [44]. The main purpose of our manuscript is to do an epidemiological systematic-review and meta-analysis on this subject in order to support (or not) the possibility of beta-thalassemia minor protection against COVID-19 (and its severe forms). In the next systematic review study, we wanted to determine the prevalence, severity, ICU admission, and mortality rates of COVID-19 in beta-thalassemic minor patients.

## 3. Review

This systematic review and meta-analysis has been conducted according to the preferred reporting items for systematic reviews and meta-analyses (PRISMA) checklist [45]. The review was registered at the International Prospective Register of Systematic Reviews (PROSPERO) with the registration number CRD42022345452.

### 3.1. Methods

Searches were conducted in PubMed (www.ncbi.nlm.nih.gov), Elsevier (www.elsevier.com) and Google Scholar (www.scholar.google.com) from 11 to 17 July 2022, to identify original research articles, reviews, and case reports that described β-thalassemic minor patients affected by COVID-19. The screening period for the studies would be from January 2020 to May 2022.

One of the authors in the team would individually extract the following data for each study (each author would check a separate database): title of the study, DOI, study design, timeline (retrospective or prospective), country, study type, age, gender, number of patients, number of patients with beta-thalassemic traits, number of COVID-19 patients, number of COVID-19 patients with beta-thalassemic trait, severity of COVID-19 (asymptomatic or mild, moderate, severe), mortality number, ICU admission, comorbidities, Diabetes mellitus, Hypertension, Cardiovascular disease, Pulmonary hypertension, Cancer, Chronic Pulmonary Disease (emphysema or/and bronchitis), Autoimmune disease (Hashimoto thyroiditis, rheumatoid arthritis, systematic lupus, and autoimmune hepatitis), Hypercholesterolemia, and Stroke. Reference lists of the included articles will be screened for additional relevant studies. Datasets will be extracted and managed in Microsoft Excel. For the MS Excel extraction sheet, we have used the Cochrane Effective Practice and Organisation of Care (EPOC) Template, modified for the needs of our study.

### 3.1.1. Risk of Bias (Quality) Assessment

The JBI (Joanna Briggs Institute) critical appraisal checklist for cross-sectional and observational studies reporting prevalence data.

### 3.1.2. Strategy for Data Synthesis

Data will be aggregated based on study design, location of reporting (country), participant demographic characteristics, etc.

Firstly, we will make a qualitative synthesis, describing the main characteristics of the studies. If possible, based on the data reported in the extracted studies, meta-analyses will be performed on each of the included parameters using Rl. Statistical significance will be set at $p$ 0.05, unless stated otherwise in the study.

Concerning first outcomes of the meta-analysis, the COVID-19 prevalence estimation will be assessed with Freeman-Tukey, the variance estimation with a DerSimonian and

Laird method, and the Cochran Q and I$^2$ method to explore statistical heterogeneity at a significance value of 0.10 [46].

Concerning the secondary outcomes (severity, ICU admission, and mortality), the relative risks/risk ratios/hazard ratios for individual studies will be combined using a random-effects meta-analysis (based on type of variables: KM or Cox regressional model). Each one will be assessed individually.

Combinations of the following medical subject headings (MeSH), terms, and keywords were used to conduct a comprehensive literature search: (beta-thalassemia or Cooley's anemia) and (SARS-CoV-2, COVID-19, or novel coronavirus). Reference lists of the eligible studies will be manually screened for the identification of additional relevant reports. The human observational studies, which reported the incidence of confirmed COVID-19 cases, mortality rate, severity assessment, or ICU admission among patients with beta-thalassemia minor and that in comparison to the general population, were included in this analysis. The language was limited to English. Studies such as case reports and review studies, and studies that did not have complete data for calculating incidences, were excluded. All inclusion and exclusion criteria were presented in the PROSPERO report (https://www.crd.york.ac.uk/prospero/display_record.php?RecordID=345452, accessed on 1 January 2023).

*3.2. Results*

The systematic search resulted in 2265 initial records, of which 871 were excluded as duplicates and 497 as irrelevant records due to the study date. 897 full-text articles were assessed for eligibility according to our inclusion criteria. Finally, 3 articles (3 studies) were found to be appropriate for quantitative synthesis [6,42,43] (Figure 2). The main characteristics of the studies are summarized in Table 1. The study designs mainly included cross-sectional methods. Publication bias was not significant according to Egger's test [47] (Table 2).

**Table 1.** Main characteristics of the included studies.

| References | 10 | 11 | 6 |
|---|---|---|---|
| Timeline | Prospective | Retrospective | Prospective |
| Duration (days) | 61 days | 181 days | 30 days |
| Country | Greece | Greece | Italy |
| Study type | Cohort study | Cohort study | Case-control |
| Total patients | 255 | 760 | 801 |
| Total patients affected by COVID-19 | 255 | 760 | 182 |
| B-thal heterozygote patients | 45 | 189 | NR |
| B-thal heterozygote patients affected by COVID-19 | 45 | 189 | 19 |
| **Severity** | | | |
| **Total patients** | | | |
| Mild affected by COVID-19 | 68 | 190 | 143 |
| Moderate affected by COVID | 113 | 373 | 39 |
| Severe and Critical affected by COVID-19 | 74 | 197 | |
| **B-thal heterozygotes** | | | |
| Mild affected by COVID-19 | 5 | 15 | 19 |
| Moderate affected by COVID | 19 | 66 | 0 |
| Severe and Critical affected by COVID-19 | 21 | 56 | |

**Table 1.** *Cont.*

| Mortality | | | | |
|---|---|---|---|---|
| Total deceased | 70 | 189 | NR | |
| B-thal heterozygote deceased | 20 | 53 | NR | |
| Total deceased by COVID-19 | 70 | 189 | NR | |
| B-thal heterozygote deceased by COVID-19 | 20 | 53 | NR | |
| **ICU admission** | | | | |
| Total ICU | 53 | NR | 39 | |
| B-thal heterozygote ICU | 11 | NR | 0 | |
| **Comorbidities in the general population** | | | | |
| Male | 153 | 448 | 70 | |
| Age: mean ± SD | 61.56 (±16.597) | 62.21 (±16.42) | 53.2 ± 18.1 | |
| Diabetes mellitus | 53 | 156 | 6 | |
| Hypertension | 142 | 420 | 27 | |
| Cardiovascular disease | 82 | 227 | NR | |
| Cancer | 29 | 85 | 4 | |
| Chronic Pulmonary Disease (emphysema or/and bronchitis) | 32 | 94 | 2 | |
| Autoimmune disease (Hashimoto thyroiditis, rheumatoid arthritis, systemic lupus, and autoimmune hepatitis) | NR | NR | 22 | |
| Hypercholesterolemia | 113 | NR | 18 | |
| Stroke | 50 | 142 | 14 | |
| **Comorbidities in B-thal minor population** | | | | |
| Male | NR | NR | NR | |
| Age: mean ± SD | NR | NR | NR | |
| Diabetes mellitus | NR | NR | NR | |
| Hypertension | NR | NR | NR | |
| Cardiovascular disease | NR | NR | NR | |
| Cancer | NR | NR | NR | |
| Chronic Pulmonary Disease (emphysema or/and bronchitis) | NR | NR | NR | |
| Autoimmune disease (Hashimoto thyroiditis, rheumatoid arthritis, systemic lupus, and autoimmune hepatitis) | NR | NR | NR | |
| Hypercholesterolemia | NR | NR | NR | |
| Stroke | NR | NR | NR | |

**Table 2.** Computed results concerning the different studied outcomes.

| | | Incidence Rate (95% CI) | Severity Rate (95% CI) | ICU Admission Rate (95% CI) | Mortality Rate (95% CI) |
|---|---|---|---|---|---|
| Overall | index | 0.9250 (0.5752; 1.4877) | 1.5933 (0.4884; 5.1981) | 0.3620 (0.0025; 51.6821) | 1.8542 (0.7819; 4.3970) |
| | z | −0.32 | 0.77 | −0.40 | 1.40 |
| | *p* value | 0.7479 | 0.4400 | 0.6881 | 0.1610 |
| I² (%) | | - | 90.0% (73.1–96.2%) | 91.7% (71.0–97.6%) | 85.1% (39.7%; 96.3%) |
| *p* value | | - | 0.0001 | 0.0005 | 0.0095 |
| Egger's test | 95% CI for bias | - | (−4.65; −0.12) | - | - |
| | *p*-value | - | 0.3138047 | - | - |

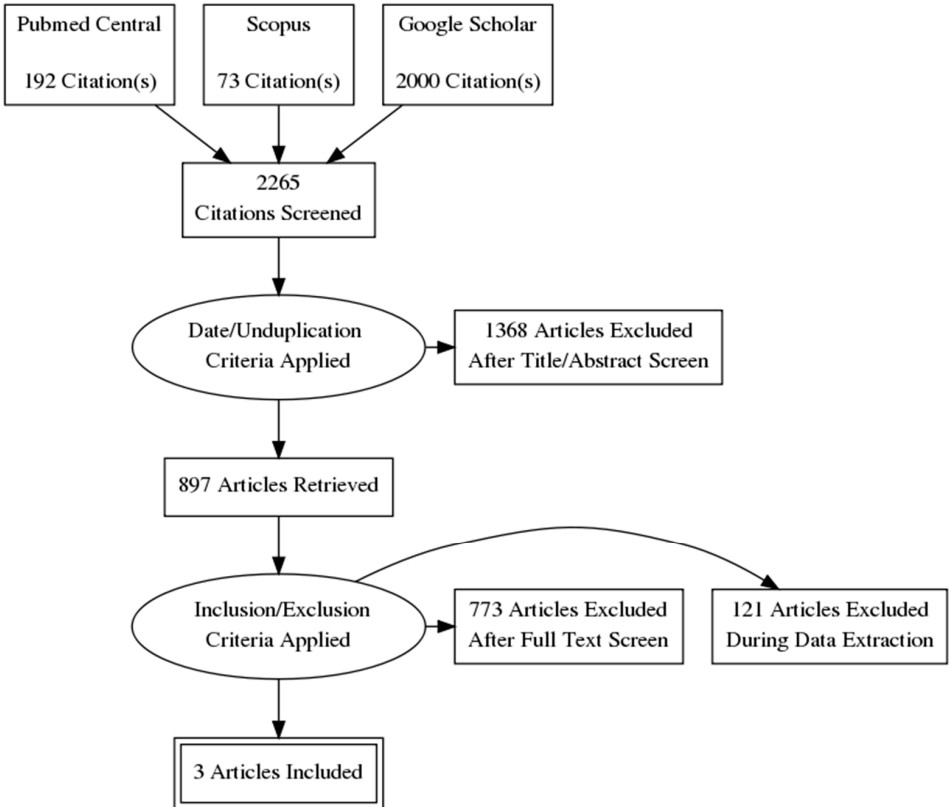

**Figure 2.** Flowchart of study identification and selection process.

Concerning the COVID-19 incidence, only one study [6] was included in the analysis of the COVID-19 incidence in beta-thalassemic carriers. Therefore, we obtained a reduction of the incidence with an Incidence Rate Ratio of 0.9250, but it was not significant with its *p*-value of 0.7479 and the trust interval [0.5752; 1.4877] (Figure 3).

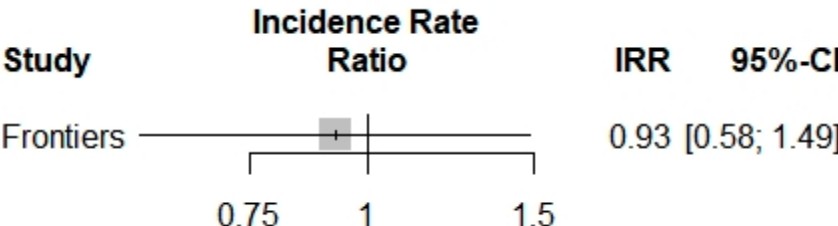

**Figure 3.** Incidence Rate Ratio of COVID-19 into beta-thalassemia carriers.

Concerning the COVID-19 severity Odds-Ratio, all 3 studies [6,42,43] were included here to compute an Odds-Ratio of 1.5933, which suggested an increase in severity in the beta-thalassemic carriers compared to the normal population. However, it stays insignificant with a 0.4400 *p*-value and a [0.4884; 5.11981] trust interval. The random-model effect was used due to the heterogeneity ($I^2$ = 90.0%) (Figure 4). Furthermore, selected studies have a large heterogeneity $I^2$ = 91.7% (71.0–97.6%) but did not indicate the presence of funnel plot asymmetry according to Egger's test (Table 2).

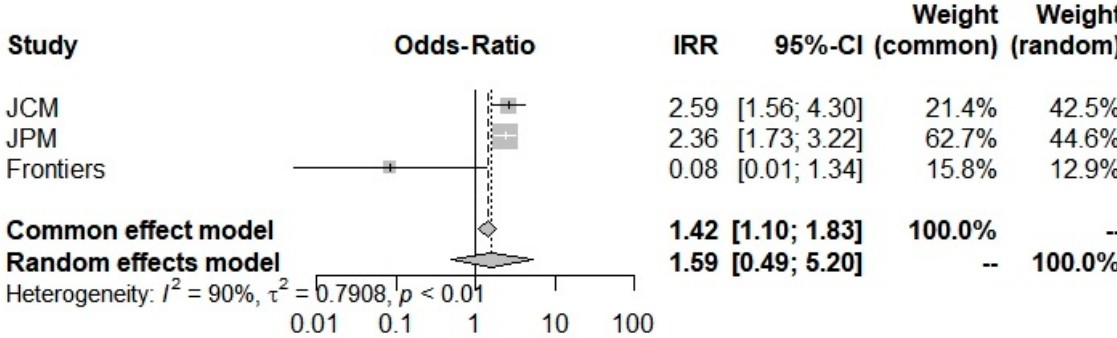

**Figure 4.** Odd-Ratios of COVID-19 severity into beta-thalassemia carriers.

Concerning ICU admission incidence, according to the random-effects model, we have computed an Incidence Rate Ratio of 0.3620 (0.0025; 51.6821) which suggests a large and insignificant decrease in COVID-19 ICU admissions concerning beta-thalassemia carriers compared to the general population (Figure 5 and Table 2). Moreover, selected studies [6,42] have a broad heterogeneity $I^2$ = 91.7% (71.0–97.6%).

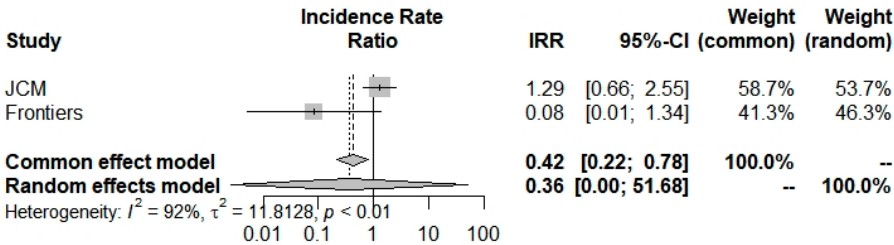

**Figure 5.** Incidence Rate Ratio of ICU admission for COVID-19 into beta-thalassemia carriers.

The last outcome was the mortality incidence rate: beta-thalassemia carriers have shown a larger but insignificant propensity (1.8542; 0.7819; 4.3970) to die with a COVID-19 infection (Figure 6 and Table 2). Selected studies [42,43] have a broad heterogeneity $I^2$ = 85.1% (39.7%; 96.3%).

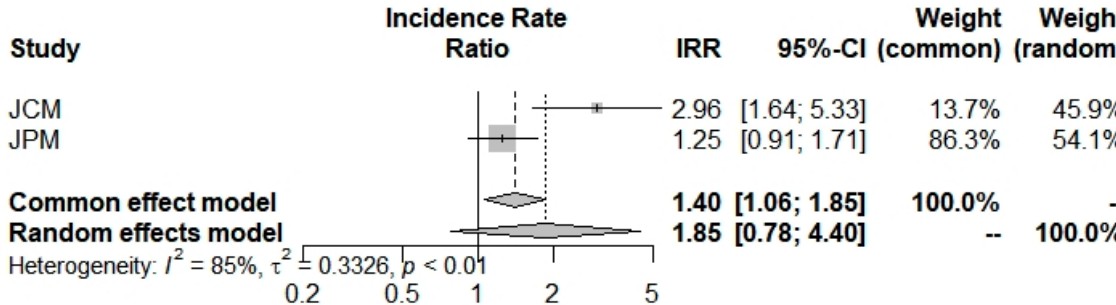

**Figure 6.** Mortality Incidence Rate Ratio for COVID-19 into beta-thalassemia carriers.

## 4. Discussion

According to our systematic-review, meta-analysis, and physiopathological hypothesis, beta-thalassemia carriers could be less affected by COVID-19 than the general population [IRR = 0.9250 (0.5752; 1.4877)], affected by COVID-19 with a worst severity [OR = 1.5933 (0.4884; 5.1981)], less admissible into the ICU [0.3620 (0.0025; 51.6821)], and more susceptible to dying from COVID-19 or one of its consequences [1.8542 (0.7819; 4.3970)]. However, all of those results remain insignificant with a bad *p*-value (respectively 0.7479, 0.4400, 0.6881, and 0.1610).

This non-significance could be mainly explained by the small number of included studies (n = 3). That is our first limitation. The second consists of the type of study; indeed, we could not assess a real incidence based on only one case-control study and two other selected studies that were COVID-19 positive cohorts. Therefore, more datasets are needed to be published in order to obtain significant incidence rate ratios and odds-ratios.

In spite of those terrible limits, this systematic-review and meta-analysis opens the way to the confirmation of a possible protection/immunity of beta-thalassemia carriers against COVID-19 concerning incidences and ICU admission markers [7,8]. However, more epidemiological studies are required in order to obtain a strong signal.

**Funding:** This research received no external funding.

**Conflicts of Interest:** The authors declare no conflict of interest.

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
