# Peer review of "Beta-Thalassemia Minor and SARS-CoV-2: Physiopathology, Prevalence, Severity, Morbidity, and Mortality"

_thalassrep, doi:10.3390/thalassrep13010003_

Round 1

Reviewer 1 Report (Previous Reviewer 2)

This is a follow up report of the Medical Hypothesis paper Lansiaux E et al of 2020. The protective effect of β-thalassaemia heterozygocity does not appear to be proven on epidemiological grounds, since statistical significance was not demonstrated. However, the analysis of the possible pathophysiological effects is very interesting and well studied. This study should be published   

Author Response

Dear reviewer 1,

Thanks for your constructive feedback. Changes were required by the other reviewer. Here is our new manuscript.

Sincere regards,

The Authors

Reviewer 2 Report (New Reviewer)

I would have liked to have found more clearly the inclusion and exclusion criteria of the selected clinical studies.

While the introduction is interesting and well argued, I did not find a clear articulation between the meta-analysis and the basic data that are discussed in it. It gives the impression that these two elements could be disjointed.
There could be more emphasis on the methodology of the meta-analysis.

Author Response

Dear Reviewer 2,

Thanks for your constructive feedback. All required changes were made into our manuscript.

Sincere regards,

The authors

This manuscript is a resubmission of an earlier submission. The following is a list of the peer review reports and author responses from that submission.

Round 1

Reviewer 1 Report

The paper is very interesting however the data showed are insufficient to a valid conclusion. Only 3 patients are included. 

They have to report more data. Is important also to eventually underline the possible difference between Th.major and Th. minor phenotypes 

Author Response

This is a careful review(and a meta-analysis)  attempting to detect differences in the response of beta thalassaemia heterozygotes to COVID-19 compared to the general population. Our results seem to indicate no statistically significant difference in incidence, severity and mortality but with a trend towards worse outcomes. We point out the limitations of our review and that more epidemiological studies are needed.  The value of our observations (based upon 3 studies) seems limited as we agree that more reports are needed, but it's a preliminary study.

Concerning the major beta-thalassemics status against COVID-19, a meta-analysis was already done on this subject (https://pubmed.ncbi.nlm.nih.gov/33317358/). Therefore, it doesn't seem useful to perform an other one without new studies published on it. Moreover, if we modify our initial inclusion and exclusion criterion, it will be a sort of PROSPERO principle violation. In this way, in absence of new published studies on this domain, an other meta-analysis upon beta-thalassemic major status will be redundant. 

In order to fullfill all your recommendations, we have performed several revisions in order to put in the light the necessity to study more patients on this specific subject and to develop this physiopathological hypothesis. Please see the attachment.

Reviewer 2 Report

This is a careful review attempting to detect differences in the response of beta thalassaemia heterozygotes to COVID-19 compared to the general population. The results seem to indicate no statistically significant difference in incidence, severity and mortality but with a trend towards worse outcomes. The authors point out the limitations of their review and that more epidemiological studies are needed.  The value of their observations seems limited as we agree that more reports are needed, but as a preliminary study it could be published 

Author Response

Dear reviewer, 

Thanks for your comments on our manuscript